# Phototoxicities Caused by Continuous Light Exposure Were Not Induced in Retinal Ganglion Cells Transduced by an Optogenetic Gene

**DOI:** 10.3390/ijms22136732

**Published:** 2021-06-23

**Authors:** Kitako Tabata, Eriko Sugano, Akito Hatakeyama, Yoshito Watanabe, Tomoya Suzuki, Taku Ozaki, Tomokazu Fukuda, Hiroshi Tomita

**Affiliations:** 1Laboratory of Visual Neuroscience, Graduate Course in Biological Sciences, Division of Science and Engineering, Iwate University, 4-3-5 Ueda, Morioka 020-8551, Iwate, Japan; ktabata@iwate-u.ac.jp (K.T.); sseriko@iwate-u.ac.jp (E.S.); g0321138@iwate-u.ac.jp (A.H.); t5518002@iwate-u.ac.jp (Y.W.); s0218017@iwate-u.ac.jp (T.S.); tozaki@iwate-u.ac.jp (T.O.); tomof009@iwate-u.ac.jp (T.F.); 2Clinical Research, Innovation and Education Center, Tohoku University Hospital, 1-1 Seiryo, Aoba, Sendai 980-8574, Miyagi, Japan

**Keywords:** gene therapy, retinitis pigmentosa, retinal ganglion cells, light damage, channelrhodopsin

## Abstract

The death of photoreceptor cells is induced by continuous light exposure. However, it is unclear whether light damage was induced in retinal ganglion cells with photosensitivity by transduction of optogenetic genes. In this study, we evaluated the phototoxicities of continuous light exposure on retinal ganglion cells after transduction of the optogenetic gene mVChR1 using an adeno-associated virus vector. Rats were exposed to continuous light for a week, and visually evoked potentials (VEPs) were recorded. The intensities of continuous light (500, 1000, 3000, and 5000 lx) increased substantially after VEP recordings. After the final recording of VEPs, retinal ganglion cells (RGCs) were retrogradely labeled with a fluorescein tracer, FluoroGold, and the number of retinal ganglion cells was counted under a fluorescent microscope. There was no significant reduction in the amplitudes of VEPs and the number of RGCs after exposure to any light intensity. These results indicated that RGCs were photosensitive after the transduction of optogenetic genes and did not induce any phototoxicity by continuous light exposure.

## 1. Introduction

Photoreceptor degeneration is induced by exposure to continuous light or UV radiation. It is also well known as light damage, and its mechanisms have been well investigated in rodents. In nocturnal animals, photoreceptor degeneration is caused by light intensity more than 2–3 times that of normal room light conditions [1]. Light-induced photoreceptor degeneration in rodents has been used as a model for human retinal degeneration arising from aging or genetic disorders such as age-related macular degeneration and retinitis pigmentosa (RP). Various factors have been reported to be involved in photoreceptor cell death by light damage. Vitamin A-deficient rats [2], rhodopsin knockout mice [3], and RPE-65 knockout mice [4] were protected against light damage, indicating that rhodopsin bleaching and its regeneration were key factors in inducing photoreceptor degeneration. Regarding the protection against retinal light damage, the use of antioxidants such as the natural L-stereoisomer of ascorbic acid [5], the free radical spin trap phenyl-N-tert-butylnitrone [6], and a TEMPOL derivative that catalyzes the degradation of superoxide [7] effectively reduces light damage. Thus, the causes and protection of phototoxicity in photoreceptor cells have been well investigated. However, there are few studies related to phototoxicities in other retinal neurons.

Recently, the application of optogenetic gene therapy to patients with RP has been explored. RP is one of the major causes of blindness worldwide. A prevalence of approximately 1 in 4000 people worldwide and various gene mutations of RP have been reported (https://www.nei.nih.gov/learn-about-eye-health/eye-conditions-and-diseases/retinitis-pigmentosa (accessed on 23 June 2021). No effective treatment for RP has yet been established. Transduction of Chlamydomonas-derived channelrhodopsin-2 (ChR2) into retinal ganglion cells (RGCs) in rd1 mice [8] and Royal College of Surgeons (RCS) rats [9] restored visual function. In addition, the effectiveness and safety of optogenetic gene therapy for RP have been investigated in behavioral studies [10,11] and immunological responses [12]. Recently, other types of channelrhodopsins, such as modified volvox-derived channelrhodopsin-1 (mVChR1) with a broad spectrum [13,14] and ChrimsonR with a red-shifted spectrum [15], have been developed, and the method of optogenetic gene therapy is becoming a promising treatment for restoring vision in blind patients [16].

RGCs are unable to receive light, except for melanopsin-expressing RGCs [17]. There is no report on RGCs induced by light damage, although secondary degeneration of RGCs is induced by photoreceptor degeneration in RCS rats [18]. By transducing the optogenetic gene into RGCs, RGCs respond to light directly without a photoreceptor-mediated pathway. The present study was designed to investigate whether photosensitive RGCs were damaged by intense light exposure.

## 2. Results

### 2.1. Patch Clamp Recordings

mVChR1 expression was clearly observed in the plasma membrane stained with wheat germ agglutinin (WGA) in HEK293 cells (Figure 1A). Photo-stimulations of LEDs at 560 nm were performed with stimulus durations of 0.1, 1.0, and 10 μW/mm^2^. Photocurrents (22.39 ± 10.44 pA) were elicited even at 0.1 μW/mm^2^ with a 10 ms duration of stimulation (Figure 1B). As the duration or intensity increased, the photocurrent increased (Figure 1C). The results of patch clamp recordings showed that photo-stimulation at 0.1 μW/mm^2^ was sufficient to depolarize the cells. The light intensities of 1 and 10 μW/mm^2^ at 560 nm corresponded to approximately 50 lx and 500 lx, respectively. Therefore, the maximum light intensity used in vivo (5000 lx) was set from 500 lx to 5000 lx.

### 2.2. Recordings of Visually Evoked Potentials

Visually evoked potentials (VEPs) were clearly recorded in the RCS rats injected with the adeno-associated virus vector (AAV)-mVChR1 after mVChR1 gene transduction (Figure 2), although the VEPs were not recordable in the only Venus-transduced retina (data not shown). As the stimulus intensities of any LED wavelength increased, the amplitudes of VEPs increased (Figure 2A). The recorded amplitudes correlated well with the stimulus intensities (Figure 2B; 465 nm: r = 0.94; 525 nm: 0.96; 640 nm: r = 0.93). No significant difference was observed in the recorded amplitudes of VEPs after any intensity of continuous light exposure (Figure 2C). The comparison of the individual data to pre-recorded amplitudes showed no significant decrease after continuous light exposure (Figure 2D).

### 2.3. Number of RGCs

The focus depth to RGCs is sometimes different when taking photographs of RGCs in flat-mounted retinas. Therefore, we took *z*-axis stacked photographs and obtained a single photograph using the extended depth of field algorithm. A single image made from 33 slices is shown in Figure 3A,B. The number of total and mVChR1-positive RGCs after a series of continuous light exposures were the same as those without light exposure (Figure 3C). Different susceptibilities of photoreceptor degeneration induced by continuous light exposure between the superior and inferior retina have been reported [19]. Therefore, we compared the number of RGCs between the superior and inferior retinas (Figure 3D). The number of total RGCs divided by the cell size in the light-exposed retina correlated well with that in the retina not exposed to light (Figure 3E).

## 3. Discussion

Optogenetic-mediated gene therapy for restoring vision is expected to be a treatment for patients with retinitis pigmentosa. We previously reported that the RP model of rats treated with ChR2- [12] or mVChR1- [13] mediated gene therapy was functionally maintained for a year, which might be a feasible treatment for RP patients. However, we did not investigate the phototoxicity of optogenetic gene-expressing RGCs against intense light. Here, we showed that both the light sensitivity of mVChR1 in HEK293 cells by patch-clamp recordings and mVChR1-transduced rats by VEPs. The intensity of 10 μW/mm^2^ at 560 nm used in patch clamp recordings was approximately 500 lx at the measured value with illuminance. Based on the lighting standard “EN12464-1 (https://www.en-standard.eu/din-en-12464-1-light-and-lighting-lighting-of-work-places-part-1-indoor-work-places/ (accessed on 23 June 2021)”, 500 lx was an appropriate light condition for normal office work. Therefore, a series of light exposures of more than 500 lx in each duration of 1 week is unusual light conditions in daily life. The results of this study demonstrated that mVChR1-expressing RGCs did not cause any phototoxicity due to continuous light exposure, which would be useful to the clinical trial of optogenetic-mediated gene therapy for restoring vision.

As shown in Figure 1, the photocurrent could be elicited at 0.1 μW/mm^2^ for 10 ms, although it was quite small. At a stimulus intensity of 1 μW/mm^2^, large photocurrents were elicited even with a duration of 10 ms. Considering that the VEP recordings used a stimulus for 10 ms and the absorption and refraction of the posterior parts such as the cornea and lens, the stimulus intensity of the VEP recording used in the in vivo study should be applied at a higher intensity than in the in vitro study. At a stimulus intensity of 20 μW/mm^2^ of 465 nm or 525 nm LEDs, large amplitudes of VEPs were clearly detected. The responses of VEPs correlated well with the stimulus intensities (Figure 2B), indicating that the stimulus intensity of 10 μW/mm^2^, which was 10 times higher than that in the in vitro study, with wavelengths of 465 nm and 525 nm, is expected to be sufficient to elicit VEPs. To investigate the effects of continuous light exposure on mVChR1-expressing RGCs, we used fluorescent light. Generally, the light intensity of the fluorescent light is measured with an illuminance meter because the fluorescent light includes various wavelengths of light. However, the unit (lux) of an illuminance meter cannot be converted to μW/mm^2^. Nonetheless, together with the stimulus intensity used in patch clamp recordings (10 μW/mm^2^ corresponded to 500 lx) and amplitudes of VEPs from the stimuli of 20 μW/mm^2^ (Figure 3C), the intensities used for the continuous light exposure were converted to normal light from unusually intense light conditions.

There are many studies related to the mechanisms of photoreceptor degeneration caused by light damage. RPE65 is an enzyme related to 11-cis retinal regeneration, and RPE65-deficient mice showed photoreceptors resistant to retinal light damage by 11-cis retinal starvation [4]. Ishizawa et al. [20] also reported that halothane, a volatile anesthetic that can compete with 11-cis retinal in opsin molecules, has a protective effect against light-induced photoreceptor degeneration. Thus, it has been reported that the chromophore, retinal, rod, and cone opsins are the factors in photoreceptor cell death caused by light. Human opsin molecules use 11-cis retinal as the chromophore, and light causes isomerization to all-trans retinal. On the other hand, channelrhodopsins, including mVChR1, belong to the microbial rhodopsin family [21], which uses all-trans retinal as a chromophore in the dark and light isomerizes all-trans retinal to 13-cis retinal. It is unlikely to show toxicity to RGCs by expressing mVChR1 because the regeneration of 11-cis retinal is unrelated to microbial rhodopsin, mVChR1. However, it has also been reported that all-trans retinal is related to photoreceptor degeneration [22]. Accumulation of all-trans retinal derivatives in RPE cells is strongly associated with the development of photoreceptor degenerative disease, Stargardt disease [23,24]. The expression of mVChR1 in retinal cells, including RGCs, might reduce trans-retinal-mediated toxicity on retinal cells using free all-trans retinal as a chromophore.

Various types of RGCs have been classified based on their morphology and function. Morphological classification of RGCs in cats [25,26] and primates [27,28] and the functional classification of RGCs in cats [29] have been reported. Regarding RGCs in rats, the classification was performed based on morphology, such as the shape and branching of the dendrites of RGCs [30,31]. It is known that the number of RGCs in RCS rats gradually decreases after photoreceptor degeneration [32,33]. We previously reported that the gradual decrease in RGCs was attenuated by transduction of the ChR2 gene [34]. On the other hand, in rd1 mice with a mutation for a rod-specific phosphodiesterase, nine different types of RGCs were especially resistant to photoreceptor degeneration and retained their dendritic geometry well beyond the complete death of photoreceptors [35]. It has also been reported that a specific cell type of RGCs is resistant to axonal injury. It was reported that the average M1 melanopsin ganglion cell somal diameter and overall morphological appearance remained unchanged in non-injured and injured retinas [36]. Thus, it is possible that a specific population of RGCs is susceptible to different types of injuries, such as photoreceptor degeneration and axonal injury. In the present study, we compared the number of RGCs and analyzed the somal size of surviving RGCs (Figure 4). The distribution of somal size of RGCs in the light-exposed retina correlated well (r^2^ = 0.987) with that in the unexposed retina (Figure 3C). More than 90% of detected RGCs were within 340 µm^2^, which almost coincides with the diameter of RGCs previously reported [31,35,37]. Therefore, the distribution of mVChR1-expressing RGCs based on cell size did not change after continuous light exposure, indicating that continuous light exposure did not cause toxicity in mVChR1-expressing cells.

## 4. Materials and Methods

### 4.1. mVChR1 Gene Transduction into HEK293 Cells

HEK293 cells (RCB1637: Riken Bioresource Center, Tsukuba, Japan) were cultured in Minimum Essential Medium (Thermo Fisher Scientific, Tokyo, Japan) supplemented with 10% fetal bovine serum under a 5% CO_2_ atmosphere at 37 °C. The plasmid vector pCAG-mVChR1 was transfected into HEK293 cells using a slight modification of the calcium phosphate method [38]. Briefly, calcium phosphate-DNA particles were added to cultured HEK293 cells, and the cells were incubated in a 5% CO_2_ atmosphere at 37 °C for 6 h. After replacing the fresh culture medium, the cells were incubated in a 5% CO_2_ atmosphere at 37 °C for 24–30 h and were then seeded onto glass slides one day before the patch clamp recordings were performed.

mVChR1 expression in HEK293 cells was investigated by staining with rhodamine WGA (RL-1022, Vector, USA) to stain the cell membrane. In detail, mVChR1-transduced cells on a chamber slide were fixed with 4% paraformaldehyde in phosphate-buffered saline (PBS; Fujifilm Wako Pure Chemical, Osaka, Japan) and incubated with rhodamine WGA (1 μg/mL) for 10 min at room temperature, followed by washing with PBS. The slide was covered with DAPI-Fluoromount-G (COSMO Bio, Tokyo, Japan) and observed under a confocal microscope (LSM-700, Carl Zeiss, Tokyo, Japan).

### 4.2. Patch Clamp Recordings

Whole-cell patch-clamp recordings were performed using an EPC-10 amplifier (HEKA Electronik, Lambrecht, Germany) under whole-cell patch clamping conditions of isolated cells, as previously described [39,40]. Photostimulations were produced using light-emitting diodes of various wavelengths (Mightex Systems, Pleasanton, CA, USA) at an intensity of 1 µW/mm^2^ to investigate the wavelength sensitivity. To measure the response to stimulus durations and intensities, the duration was set to 10, 100, and 1000 ms, and the intensity of light was adjusted to 0.1, 1, and 10 µW/mm^2^ at 560 nm of LED by setting an appropriate current. The intensity of light was measured using an optical power meter (Hioki, Nagano, Japan) and an illuminance meter (Mother Toll, Nagano, Japan). The patch clamp data were collected by filtering at 10 kHz and sampling at 5 kHz. The internal solution contained 130 mM CsCl, 10 mM HEPES, 2 mM MgCl_2_, 0.1 mM CaCl_2_, 10 mM NaCl, 2 mM Na_2_ATP, and 1.1 mM EGTA, with the pH adjusted to 7.2 for whole-cell current recordings. The external solution contained 138 mM NaCl, 3 mM KCl, 1 mM CaCl_2_, 2 mM MgCl_2_, 4 mM NaOH, and 10 mM HEPES, with the pH adjusted to 7.4 by HCl.

### 4.3. Animals

All animal experiments were conducted in accordance with the guidelines of the Animal Experiment Committee of Iwate University, Japan. RCS rats were obtained from CLEA Japan (Tokyo) and were housed under conditions of cyclic light (on: 8:00 am, 200 lx; Off: 8:00 pm) at 23 ± 1 °C and provided ad libitum access to laboratory chow and water. In the present study, 9-month-old male RCS rats (*n* = 8) were used.

### 4.4. Preparation of the AAV Vector

Plasmid vectors containing mVChR1 [14] were used to produce AAV vectors. The AAV Helper-Free System (Agilent Technologies, Santa Clara, CA, USA) was used to produce infectious AAV virions, according to the manufacturer’s instructions. AAV vectors were purified using a previously described single-step column purification method [41]. The concentration of the purified AAV vectors was determined by measuring the levels of the AAV capsid protein using ELISA (PROGEN, Heidelberg, Germany).

### 4.5. Intravitreal Injection of AAV

Rats were anesthetized by intramuscular injection of 45 mg·kg^−1^ ketamine and 4.5 mg·kg^−1^ xylazine. Using an operating microscope, 5 μL of a suspension containing 1 × 10^12^ virions·mL^−1^ was intravitreally injected through the ora serrata using an automatic syringe (Neurosyringe AC, ACrux Inc., Morioka, Japan) with a 32-gauge needle (Hamilton Company, Reno, NV, USA).

### 4.6. Light Exposure

Rats were housed under normal conditions of cyclic light for 8 months after AAV injection, were placed in a light box (041001, NK system, Osaka, Japan) adjusted to the light intensity at 500 lx, and were kept for a week. VEPs were recorded 1 day after continuous light exposure. After recording VEPs, rats were returned to the light box, kept for a week under normal conditions of a cyclic light as a recovery period, then exposed to the next continuous light condition. VEPs were recorded after sequentially increasing the light intensity (Figure 4).

### 4.7. Recording VEPs

VEPs were recorded as previously described [13]. Briefly, at least 7 days before the recordings, a recording and reference electrode were implanted epidurally. For the recording of VEPs, the rats were anesthetized with intramuscular ketamine (45 mg/kg) and xylazine (4.5 mg/kg), then their pupils were dilated with 0.5% tropicamide and 0.5% phenylephrine hydrochloride. VEPs were recorded using PuREC (Mayo Corp., Aichi, Japan). Photic stimuli of various intensities were generated using LEDs with different colors (wavelength: 465, 525, and 640 nm; VE-LED, ACrux Inc, Morioka, Japan) and applied for 10 ms at 1 Hz. The high- and low-pass filters were set at 500 Hz and 3 Hz, respectively. VEP responses were consecutively measured 200 times, and the response waveforms were averaged.

### 4.8. Preparations of Whole Mounted Retina and Retinal Section

After the final recording of VEPs (5000 lx exposure), retrograde labeling of RGCs was performed as previously described [9]. Labeling was performed by injecting 4 μL of 2% aqueous fluorogold (Fluoro-Gold, Fluorochrome, Englewood, CO, USA) containing 1% dimethylsulfoxide into the superior colliculus using a Hamilton syringe with a 32-gauge needle. Seven days after labeling, rat eyes were fixed overnight at 4 °C in 4% paraformaldehyde prepared in phosphate-buffered saline (PBS; Fujifilm Wako Pure Chemical, Osaka, Japan). After rinsing with PBS, the posterior parts such as the cornea, iris, and lens were removed and the retina was detached from the eye cup. The specimen was flat-mounted on a slide and embedded in a mounting medium (Vectashield, Funakoshi, Tokyo, Japan). The flat-mounted retinas were observed and photographed using a fluorescence microscope (Keyence, BZ-9000, Osaka, Japan). Three areas in each quadrant of a whole mounted retina, totaling 12 areas, were photographed with an interval of *z*-axis at 0.3 μm, with a total of 33 slices. To calculate the number of RGCs, a single image was created from 33 images using the extended depth of field algorithm (full focus) in BZ-9000 software (Keyence, Tokyo).

### 4.9. Statistical Analysis

Statistical analyses for the in vitro experiments were performed using GraphPad Prism (MDF, Tokyo, Japan). Tukey’s multiple comparison test and Dunnett’s multiple comparison test were used for statistical analyses.

## Figures and Tables

**Figure 1 ijms-22-06732-f001:**
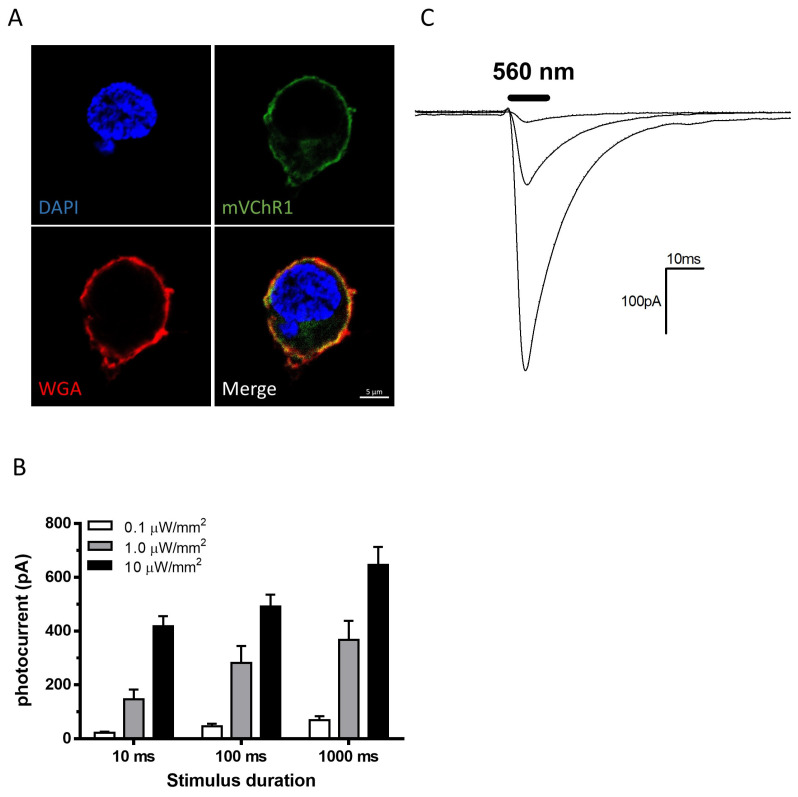
Photocurrents elicited with different intensities and durations. mVChR1 expression was observed as Venus fluorescence (green), while the plasma membrane and nuclei were stained with WGA (red) and DAPI (blue), respectively (**A**). mVChR1 expression was clearly localized in the plasma membrane. Photocurrents were recorded by stimuli with 560 nm LEDs at various intensities and durations ((**B**), *n* = 8). Typical waveforms were recorded with a stimulus at 0.1, 1.0, and 10 µW/mm^2^ for 10 ms (**C**). mVChR1, modified volvox-derived channelrhodopsin-1.

**Figure 2 ijms-22-06732-f002:**
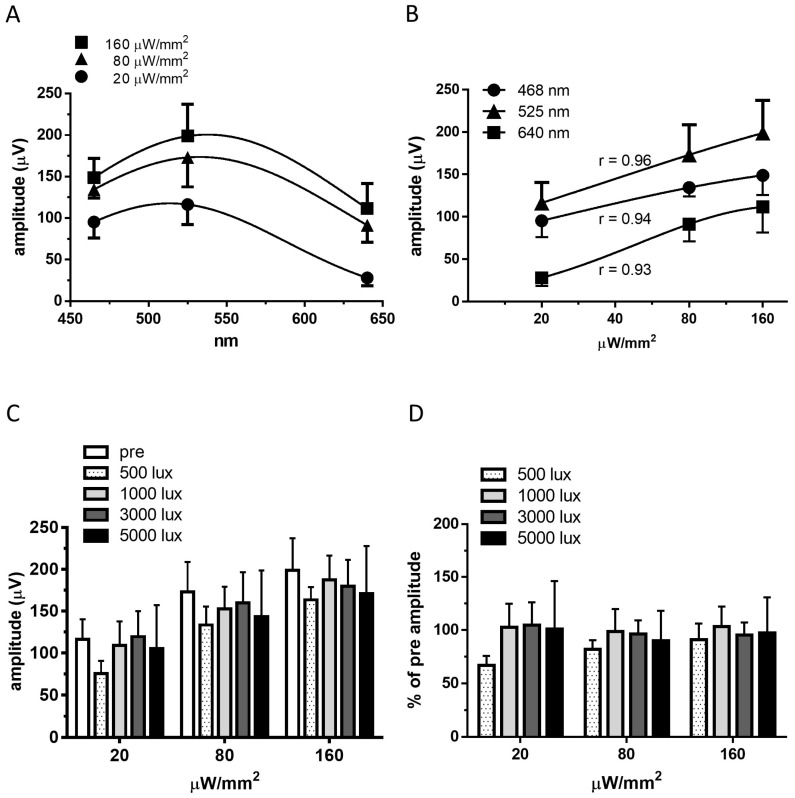
Visually evoked potentials were recorded in RCS rats with transduced mVChR1 gene. The wavelength sensitivities (**A**) and stimulus intensity responses (**B**) of the VEPs in mVChR1 transduced rats (mean ± S.E., *n* = 4). The relationships between amplitudes and stimulus intensities correlated well. Values of Pearson’s r are shown in (**B**). Recorded amplitudes of the VEPs with a stimulus of 525 nm LED (**C**) and individual percentages compared to the pre-recorded values (**D**) after a series of continuous light exposures. mVChR1, modified volvox-derived channelrhodopsin-1; RCS, Royal College of Surgeons; VEP, visually evoked potential.

**Figure 3 ijms-22-06732-f003:**
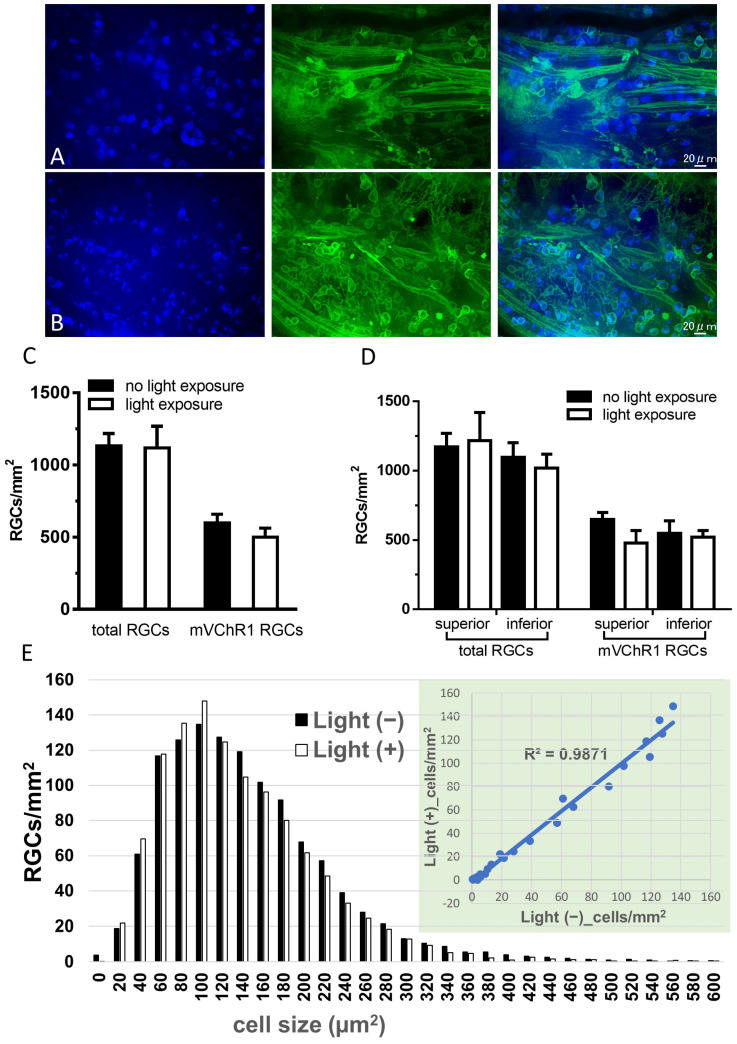
Comparison of the number of RGCs after continuous light exposure. Retrograde labeling of RGCs (blue), mVChR1-expressing cells (green), and merged photographs are shown in (**A**) (no light exposure) and (**B**) (continuous light exposure). The number of total and mVChR1-positive RGCs was counted (**C**). The RGCs were counted by dividing them into superior and inferior parts (**D**). Cell sizes in the retina without light exposure and the retina with continuous light exposure were analyzed (**E**); both cell sizes were well correlated (r^2^ = 0.9871). mVChR1, modified volvox-derived channelrhodopsin-1; RGC, retinal ganglion cells.

**Figure 4 ijms-22-06732-f004:**
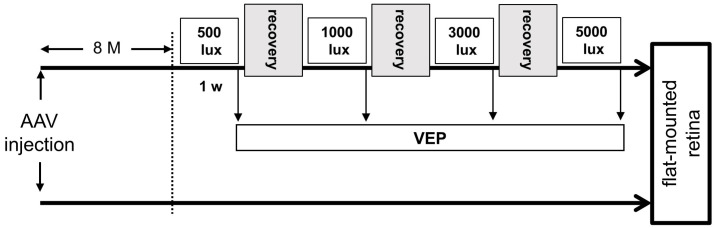
Experimental design for evaluating the phototoxicities on mVChR1-expressing RGCs.

## Data Availability

The datasets used and/or analyzed during the current study are available from the corresponding author upon reasonable request.

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
