# Peer review of "Phototoxicities Caused by Continuous Light Exposure Were Not Induced in Retinal Ganglion Cells Transduced by an Optogenetic Gene"

_ijms, 2021, doi:10.3390/ijms22136732_

Round 1
Reviewer 1 Report
The manuscript entitled "Phototoxicities caused by continuous light exposure were not induced in retinal ganglion cells transduced by an optogenetic gene" by Tabata and colleagues addresses the question of whether or not transduced RGC would become susceptible to light damage. They test light intensities to be used in transduced fibroblasts prior to exploring the situation in transduced rats. The text is well written on the whole, the illustrations mostly of good quality, and the methods described in adequate detail. My comments are attached below.
This reviewer has no problem with the experiments themselves, which seem to have been performed correctly. My main comment is: why? I see no scientific reason to believe that channel rhodopsin-transduced cells would be vulnerable to light damage. The authors basically say it themselves, there is no evidence of retinal cell damage outside of rods (maybe cones) under light stress conditions, and there have been no reports of channel rhodopsin leading to increased cell death. The mechanism of action of such proteins is completely different from mammalian rhodopsin, which has been incriminated in the pathology (as the authors rightly say). The conjecture about superior vs inferior retina is relevant for photoreceptors but not for RGC. They talk a little about potential effects of all trans retinal, which is quite toxic, but if I understand their arguments they seem to suggest the gene transfer would protect against such toxicity? As far as I can see, the only value in this study is to formally preclude that such techniques might induce cell death, which since they have clinical interest is an important fact to establish. In contrast, one could interpret the fact that RGC loss does occur secondarily to photoreceptor loss in retinal degeneration to actually limit the usefulness of such optogenetic approaches, since the transduced target cell would die anyway. But I remain very sceptical about the rationale.
One minor methodological question: in the image showing transduced and retrogradely labelled RGC (Figure 3), it looks like about half the RGC carry the transgene, but it is difficult to really see this from the co-localization data. Could this be improved or modified to indicate cells in each population?
Author Response
Dear reviewer 1,
Thank you for your detailed review and giving us suggestions.
A kind through explanation is very helpful for us to revise our manuscript.
I revised our manuscript following your comments.
I appreciate it if you re-review our manuscript.
Comment
One minor methodological question: in the image showing transduced and retrogradely labelled RGC (Figure 3), it looks like about half the RGC carry the transgene, but it is difficult to really see this from the co-localization data. Could this be improved or modified to indicate cells in each population?
RE: Modified the graph. Line 110.
Reviewer 2 Report
The manuscript by Tabata et al entitled “Phototoxicities caused by continuous light exposure were not induced in retinal ganglion cells transduced by an optogenetic gene” reports analysis of the influence of the exposure to continuous light on visually evoked potentials and the number of retinal ganglion cells transduced with modified volvox-derived channelrhodopsin-1 in the rat retina. This study is particularly relevant in view of a recent publication of the first case of partial recovery of vision in a human patient using a similar approach [PMID: 34031601]. I recommend the manuscript by Tabata et al for publication in the International Journal of Molecular Sciences after a minor revision that should address the issues listed below.
Line 19: The abbreviation “RGS” is used in the Abstract, but not explained.
Line 55: “ChrimsonR with a red-shifted spectrum [15]”. Please also cite the above-mentioned paper [PMID: 34031601], in which ChrimsonR was used to partially restore vision in a human patient.
Line 75: The X axis label in Figure 1B should be “Stimulus duration”, not “Wavelength”.
Line 80: The legend of Figure 1C says that all three traces were recorded at 0.1 μW mm-2. From their amplitudes and a description in the text it is clear that they were recorded at 0.1, 1 and 10 μW – please correct the legend.
Lines 87-88: “The stimulus intensities correlated well with the recorded amplitudes…” The stimulus intensity was the condition set by the experimenters, and the VEP amplitude was the measured response. Therefore, it is the amplitude that correlated with the intensity, not the other way around.
Lines 129-130: “Therefore, a series of light exposure of more than 500 lx in each duration of 1 week is unusual under intense light conditions.” I failed to understand what the Authors meant by this sentence – please rephrase.
Author Response
Dear reviewer 2,
Thank you for your detailed review and giving us suggestions.
A kind through explanation is very helpful for us to revise our manuscript.
I revised our manuscript following your comments.
I appreciate it if you re-review our manuscript.
Comments and responses
Line 19: The abbreviation “RGS” is used in the Abstract, but not explained.
RE: Abbreviated RGCs. Line19: retinal ganglion cells (RGCs)
Line 55: “ChrimsonR with a red-shifted spectrum [15]”. Please also cite the above-mentioned paper [PMID: 34031601], in which ChrimsonR was used to partially restore vision in a human patient.
RE: Added the ref. Line 57; [16]
Line 75: The X axis label in Figure 1B should be “Stimulus duration”, not “Wavelength”.
RE: Corrected the X axis label. Line 75
Line 80: The legend of Figure 1C says that all three traces were recorded at 0.1 μW mm-2. From their amplitudes and a description in the text it is clear that they were recorded at 0.1, 1 and 10 μW – please correct the legend.
RE: Corrected the legend. Line 80
Lines 87-88: “The stimulus intensities correlated well with the recorded amplitudes…” The stimulus intensity was the condition set by the experimenters, and the VEP amplitude was the measured response. Therefore, it is the amplitude that correlated with the intensity, not the other way around.
RE: Corrected the sentence. Line 87-88; “The recorded amplitudes correlated well with the stimulus intensities”
Lines 129-130: “Therefore, a series of light exposure of more than 500 lx in each duration of 1 week is unusual under intense light conditions.” I failed to understand what the Authors meant by this sentence – please rephrase.
RE: Rewrote the sentence. Line 129-130; Therefore, a series of light exposure of more than 500 lx in each duration of 1 week is un-usual light conditions in a daily life.